# Analysis of Research Trends on Elbow Pain in Overhead Sports: A Bibliometric Study Based on Web of Science Database and VOSviewer

**DOI:** 10.3390/healthcare10112242

**Published:** 2022-11-09

**Authors:** Wei Han Li, Maryam Hadizadeh, Ashril Yusof, Mohamed Nashrudin Naharudin

**Affiliations:** Faculty of Sports and Exercise Science, University of Malaya, Kuala Lumpur 50603, Malaysia

**Keywords:** elbow pain, overhead sports, bibliometric study, VOSviewer

## Abstract

The publications on elbow pain (EP) in overhead sports are increasing. The results of previous studies mostly focus on the influence of EP in the ball game and throwing sports. Thus, a bibliometric analysis of these publications may show the direction of hot topics and future research trends. The purpose of this study is to identify the research trends on EP in overhead sports. For the methods, the first step is to use the main keywords of ‘Elbow pain’ and ‘Overhead sport’ merging auxiliary vocabulary to reach the relevant global publications between 1970 and 2022 in the Web of Science (WoS) database. The literature data set is imported into EndNote literature manager software to remove duplication. Secondly, the duplication-reduced articles are imported to an Excel sheet according to the inclusion and exclusion criteria of this study. In the third step, VOSviewer software is applied as the main analysis tool in extracting data for analysis from the articles. Then, the main research results for three aspects are obtained by VOSviewer software which extracted and analyzed the parameters of author name, article title, publication journal, keywords, organization, publication country/region, and the sum of times cited from 455 qualified papers. The study found that the United States of America made the most outstanding contribution to this theme study. The research on EP in overhead sports in China requires more attention from scholars. EP in swimming is a new research direction worthy of attention. In conclusion, the research results prove the growth trend of EP in overhead sports. The EP problem not only exists in the ball game and throwing sports but also swimming. Sport commercialization and the involvement of related professional sports organizations determine the degree of EP’s attention in a specific sport and the development of solutions. The development of a region or country also affects the depth and scope of EP study. Clinical research development and in-depth exploration are one of the bases to solve EP problems. Non-clinical action is beneficial to EP patients, but it still needs to be explored and studied.

## 1. Introduction

Elbow pain (EP) is one of the most reported symptoms of upper limb joint discomfort in various overhead sports [1,2,3]. It usually causes soreness in arm muscles [4]. If the patient neglects EP or continues exercise and competition with discomforts, it may lead to various inflammations such as tendonitis, ‘tennis elbow’, or ‘golf elbow’ [5,6,7,8]. At present, most research relevant to EP in sport science focuses on the trend in ball sports [2]. Other types of sports have been proven to have EP issues, but the attention is less [9]. Prevalence assessment of risk factors and effective solutions in EP-relevant research is the direction of current research interest [1,6,10]. In addition, the current research trend on EP has led most scholars to focus on ball games and throwing sports, which produced less attention for EP studies in other sports [1,11,12].

The most concerning overhead sports associated with EP are baseball, handball, tennis, golf, softball, and table tennis [2]. The incidence and prevalence rates of overuse injuries of the elbow are highest in baseball (58%), followed by tennis (35%), and softball (17%) [2]. However, scholars have recently started to explore this issue in various other sports after scientists realized that the EP issue has not received enough attention. One study evaluated that the risk of EP is abundant among golf players [13]. Another study found that EP is most common among gymnastics and rock climbing, with proportions of 10–83% and 5–58%, respectively [11], where EP has been shown to affect sports performance through specific injury and discomfort positions in youth gymnasts [14]. In addition, EP represents a 20.5% prevalence rate for the elbow with a shoulder injury in synchronized swimming athletes [9]. Furthermore, a recent study discovered that 2.2% of swimmers suffered from EP in the US National Collegiate Athletic Association [4]. Therefore, it is purported to be important to study the trends of previous publications on EP for gaining a better perspective on these sports related issue.

The research on EP is no longer limited to ball games in overhead sports, but it is still insufficient in other types of overhead sports. A recent study clearly expressed that the existing research on the shoulder and elbow evaluation of elite swimmers is seriously insufficient and cannot establish the correlation between the use of certain clinical objective indicators and the development of injury [15]. The health of gymnasts’ elbow joints needs the attention of coaches and professionals in preventive treatment, training methods and loads, and rehabilitation [14]. However, the existing research seems unable to guide scholars provide a meaningful overview of the abovementioned issues. Therefore, a visual bibliometric analysis can help scholars see the research trends for these issues by analyzing the existing literature related to EP. VOSviewer is seen as one of the best software products for the visual analysis of scientific literature bibliometric and knowledge networks. Through this method, scientists are able to study research status, hot spots, and trends through unique viewpoints including clustering, superposition, density, and frequency, and study a certain field from multiple angles [16]. In recent years, scholars began to pay attention to various sports injuries by using VOSviewer analysis, such as for the anterior cruciate ligament of the knee [17], brain biomechanics [18], and recreational female footballers [19]. However, to the best of our knowledge, there is no relevant bibliometric research on EP in overhead sports.

To sum up, the previous researches which focused on EP no longer satisfy the requirements of the current study development. Finding out the research trends for EP will be the first step to do more research on EP, especially in overhead sports. Therefore, the purpose of this study was to identify the research trends for elbow pain in overhead sports. The main outcomes of this research will not only assist related professionals and practitioners to reduce EP risks and support the formulation of suggestions to improve sports performance, but also will offer potential and specific study spots for future research direction.

## 2. Materials and Methods

### 2.1. Search Strategy

Articles on elbow pain in overhead sports were obtained from the Web of Science (WoS) (https://www.webofscience.com/) database which accessed on 2 April 2021. It included Science Citation Index Expanded (SCI-Expanded), Emerging Sources Citation Index (ESCI), Social Sciences Citation Index (SSCI), Book Citation Index-Science (BKCI-S), and Conference Proceedings Citation Index-Science (CPCI-S). The period covered was from 1970 to 2022, and the final search was performed on 10 August 2022. In this study, both formulas of ‘TS = Topic’ and ‘TI = Title’ were used as the search strategy. Details of searching formulas and merging auxiliary words are shown in Appendix A.

### 2.2. Article Selection

All articles were first imported to Endnote (UMLibrary-PTM-asas 3092075999, EndNote 20, Bld 14672) for deduplication. The researcher (WH) screened all articles by the title and abstract independently before applying the following inclusion criteria: *study population* (overhead athletes including <40 years old, with pain or overuse/chronic injury of the elbow, shoulder, forearm muscle, or upper extremity muscle), *study design* (prospective or retrospective cohort studies, case-control studies, or cross-sectional studies), and *outcomes* (incidence and/or prevalence of reported pain or overuse injuries or risk factors for obtaining pain or overuse injuries), *Web of Science Categories* (Sports science). The exclusion criteria were *study participants* (Paralympic athletes), and *document types* (Discussions or Meeting Abstracts or Letters or Corrections or Discussions or Notes or Early Access or Proceedings Papers or Editorial Materials).

No restrictions were applied to the publication date. The search language used was English. Review articles were also included. Reference lists of the eligible articles were checked for any additional relevant articles.

### 2.3. Data Extraction

When article selection was completed, the following data were extracted for visualized and bibliometric analysis that included author name, article title, publication journal, keywords, organization, publication country/region, and the sum of times cited.

### 2.4. Visualized and Bibliometric Analysis (VOSviewer)

VOSviewer is a software used to construct and visualize bibliometric networks. These networks may, for instance, include journals, researchers, or individual publications, and they are constructed based on citation, bibliographic coupling, co-citation, or co-authorship relations [16]. VOSviewer also offers text mining functionality that can be used to capture co-occurrence networks of important terms extracted from a body of scientific literature [20].

In this study, VOSviewer (version 1.6.17) was used to visualize keyword co-occurrence, citations and publications, bibliographic coupling in country and sources, and themes and trend topics. Two standard weight attributes were applied which were defined as the “Links attribute” and the “Total link strength attribute” [21,22].

Both files of ‘TS = Topic’ and ‘TI = Title’ were exported using Endnote software following Text File (*.txt) format with APA 6th output style. VOSviewer created a map based on bibliographic data (Text File of ‘TS = Topic’ and ‘TI = Title’) for the visualized analysis.

## 3. Results

### 3.1. Overview of Documents Output

A total of 2745 documents on the theme of elbow pain were identified in the WoS from the years 1984–2022. A total of 1902 documents on the topic and title of elbow pain in overhead sports were screened in the WoS database, which included 1666 (91.8%) original research articles and 149 (8.21%) review articles, 50 (2.76%) early access, 29 (1.6%) proceedings papers, and 8 (0.44%) book chapters. The screened total included 1815 publications that were written in English. Based on our inclusion and exclusion criteria, 455 eligible papers were analyzed by VOSviewer (Figure 1).

### 3.2. Trend Analysis of the Citations and Publications

We obtained 455 English publications regarding EP in overhead sports. They were published from the year 1984 to 2022 and have been cited 11,546 times in total with an average of 25.38 citations per publication. In recent years, the number of articles and citations has generally appeared in a positive-growth trend (Figure 2). The peak publication year was 2018 (39 papers) with citations 820 times. The peak citation year was in 2021 (1600 times). One hundred eighty-three papers were published in the past five years (the years 2017 to 2022), accounting for 40.22% of the total, approximately half the total number of 455 publications.

The top ten journals, organizations, authors, and countries for ‘elbow pain’ are summarized in Table 1a–d. The highest number of documents was 50 in the *Journal of Shoulder and Elbow Surgery* with a count of citations of 1011. The 31 papers published in the *American Journal of Sports Medicine* had the peak citations number at 2477 in the top ten journals list for ‘elbow pain’. Although the 12 studies published in *Medicine and Science in Sports & Exercise* placed 10th, its count of citations reached 1218, which claimed third place (Table 1a).

The American Sports Medicine Institute ranked first in Table 1b with 10 documents and a total citations number of 1668. The Norwegian School of Sports Science closely followed in second place with nine publications and 594 citations.

Table 1c shows the top ten authors of ‘elbow pain’ publications. The number of articles from the 1st to the 10th ranged from 7 to 5. ANDREWS, JAMES R was not only in first place in number of publications but also obtained 1181 citations. FLEISIG, GLENN contributed five publications, but his citations at 872 ranked second place

In Table 1d of the top ten countries of ‘elbow pain’ publications, the USA with 226 articles and 6530 citations is in the first place without dispute. For the second to tenth places, the number of publications ranged from 25 to 11. Norway contributed 18 publications, but its citations were 698, which ranked second place.

### 3.3. Trend Analysis of Keywords Co-Occurrence

Keywords collected from the authors of the articles which occurred more than five times in the WoS core database were enrolled in the final analysis. There were 1710 keywords and 174 met the threshold. The keywords that occurred most were ‘elbow’ (total link strength 537) and ‘shoulder’ (total link strength 429), which had strong links with ‘biomechanics’, ‘baseball’, ‘ulnar collateral ligament’, ‘risk-factors’, ‘pain’, ‘epidemiology’, and ‘performance’. The total link strength of each was more than 210 (Figure 3). A Density Visualization was also created to show the frequency of keywords that occurred more than five times. It indicated that ‘elbow’ was the most frequent, followed by ‘shoulder’ and ‘biomechanics’ (Figure 4).

Four clusters of elbow pain studies are presented in Figure 3. The red cluster involved baseball players, tennis elbow, and management. The green cluster involved shoulder and elbow motion in biomechanics. The blue cluster involved sports epidemiology. The yellow cluster involved swimming performance related to the elbow. Including 174 items, the total link strength was 6288 among the four clusters. The strongest link of the clusters is the keyword ‘elbow’ in red, which has 537 total link strength with 83 occurrences. The keyword ‘shoulder’ ranked second with 429 total link strength and 62 occurrences. The keyword ‘performance’ ranked third with total link strength 215 accompanied by 41 occurrences. The keyword ‘swimming’ appeared to have close links with the keyword ‘performance’, which was 77 total link strength with 22 occurrences. The fourth cluster, blue, has the keyword ‘epidemiology’ which occurs with total link strength of 229 and 42 occurrences, and it was linked with the keyword ‘prevalence’ with 81 occurrences and 229 total link strength.

### 3.4. Trend Analysis of Bibliographic Coupling of Countries and Sources

Figure 5a shows bibliographic coupling of countries is clustered and 20 met the threshold out of 43 countries. The minimum number of documents for a country was five. It includes the USA, Japan, England, Norway, Netherlands, Germany, Canada, Belgium, France, Australia, Italy, Switzerland, Spain, Finland, Denmark, South Korea, Sweden, New Zealand, Portugal, and the People’s Republic of China.

The USA was the country that paid the earliest attention to elbow pain and produced the most publications. The total link strength was 12,067 with 226 documents. Its average publication year was around the year 2011. The countries having strong links with the United States are Japan, England, Germany, and the Netherlands. Their total link strength followed it with 5668, 3444, 1410, and 1782, respectively. Their average publication years were between 2014 and 2017.

However, the People’s Republic of China, Spain, and Italy are the latest countries to be concerned about elbow pain. Their average publication years followed in 2018, 2018, and 2019, respectively. Their total link strengths were 230, 790, and 839 (Figure 5a).

Figure 5b show the 25 journals that met the threshold from the total of 75 sources. The minimum number of documents for a source was five. They include the *American Journal of Sports Medicine*, *Clinics in Sports Medicine*, *Orthopedic Journal of Sports Medicine*, *Journal of Shoulder and Elbow Surgery*, *Sports Health—A Multidisciplinary Approach*, *Journal of Athletic Training*, *Sports Medicine and Arthroscopy Review*, *Current Sports Medicine Reports*, *International Journal of Sports Physical Therapy*, *British Journal of Sports Medicine*, *Physician and Sports Medicine*, *Sports Medicine*, *Knee Surgery Sports Traumatology Arthroscopy*, *Medicine and Science in Sports and Exercise*, *Journal of Science and Medicine in Sport*, *International Journal of Sports Medicine*, *Journal of Strength and Conditioning Research*, *Journal of Sports Medicine and Physical Fitness*, *Journal of Sports Sciences*, *Journal of Orthopedic Trauma*, *Scandinavian Journal of Medicine & Science in Sports*, *BMJ Open Sport & Exercise Medicine*, *European Journal of Sports Science*, and *Arthroscopy—The Journal of Arthroscopic and Related Surgery*.

The *Journal of Shoulder and Elbow*, *Clinics in Sports Medicine*, *American Journal of Sports Medicine*, *Orthopedic Journal of Sports*, and *Sports Health—A Multidisciplinary Approach* have the strongest total link strengths at 2880, 3243, 4472, 3160, and 2663, respectively. The oldest average publication year is the year 2000 for *Clinics in Sports Medicine*.

Furthermore, the *Orthopedic Journal of Sports*, *Sports Health—A Multidisciplinary Approach*, *BMJ Open Sport & Exercise Medicine*, and the *International Journal of Sports Physical Therapy* are the most recent journals focusing on elbow pain. Their average publication years were all later than the year 2018 (Figure 5b).

### 3.5. Trend Analysis of Themes and Topics

Figure 6a shows the network map of the topics according to keywords applied from 2010 to 2018. The major occurrences of ‘elbow’, ‘shoulder’, ‘biomechanics’, ‘epidemiology’, and ‘performance’ represented by green are concentrated in the average year of 2014. New research events represented by light green, such as ‘swimming’, ‘exercise’, ‘prevalence’, ‘injuries’, and so on, appeared around 2016. The latest research topics represented by yellow, such as ‘knowledge’, ‘pitch count’, ‘risk factor’, and so on, occurred after 2018. Other colors for research events indicate the years before 2012, or even older.

Figure 6b,c shows that the trending topics of ‘prevalence’ and ‘swimming’ applied from 2010 to 2018. The topic of ‘prevalence’ with total link strength of 81 and occurrences of 16 appeared after the year 2016. It is strongly linked with keywords ‘elbow’, ‘shoulder’, ‘exercise’, and ‘epidemiology’. The latest keywords linked to the topic of ‘prevalence’ are ‘collateral ligament reconstruction’, ‘injury prevention’, ‘illness’, and ‘adolescence’, and they occurred after 2018. The keywords ‘performance’, ‘pain’, ‘electromyography’, ‘exercise’, ‘adaptations’, ‘velocity’, ‘distance’, and ‘gender’ are closely linked with the topic of ‘swimming’ in the average year 2014. Both the latest keywords of ‘swimming performance’ and ‘elite swimmers’ are included, and they appear after the year 2018. All indicators in the four figures show the most recent publications from green to yellow. More studies focusing on ‘knowledge’, ‘pitch count’, ‘risk factor’, ‘elite swimming’, and the ‘prevalence’ of matters of the elbow have been published recently.

## 4. Discussion

To our knowledge, this is the first bibliometric analysis of publications pertaining to EP in overhead sports that has ever been conducted. We aimed to obtain an all-around overview and trend analysis of subject research in this field.

The overview results show that the cumulative amount of publications gradually increased in the past 10 years after the first research on EP in overhead sports appeared in 1984. Notably, in the past five years (2017–2021), the number of articles published on this topic is close to half of the total publications. One of the possible reasons for this is that the ‘Munich Consensus Statement—Terminology and Classification of Muscle Injuries in Sport’ was formulated to systematically judge athletes’ specific physical injuries with the support of the IOC (International Olympic Committee), FIFA (Fédération Internationale de Football Association), and UEFA (Union of European Football Associations) in 2012 [23]. The top ten rankings (journals, institutions, authors, and countries) show that institutions, universities, and journals in the United States have made great contributions to solving EP in baseball. This may be influenced by the commercialization of MLB (Major League Baseball) in the US. High-density events lead to an increase in the probability of athletes suffering from EP, while the benefits from commercial competitions speed up the solution of EP problems [24]. Thus, the contribution of the United States to EP is worthy of reference for potential future research.

The results of keyword co-occurrence analysis assist researchers in accurately catching sight of the trends of specific research elements on EP. Four colors clustering represents the research trends in EP-related studies. The red cluster shows that researchers pay more attention to the influence of clinical medical theory and surgical methods for EP solutions, so the keywords ’ulnar collateral ligament’, ’reconstruction’, and ’tennis elbow’ show a strong connection. The factors of ‘shoulder’ and ‘biomechanics’ are the main concern of researchers in the green cluster, therefore, important for researchers to consider when exploring EP related problems. The blue cluster mainly involves the evaluation of facts. Scholars focus on ‘epidemiology’, ’risk factors’, ’precaution’, and ‘shoulder pain’. A possible inference is that the reasons for the impact of EP in overhead sports are still in a state of having insufficient research evidence [25]. The last cluster (yellow) clearly shows the perspective of researchers in consideration of how EP affects sports performance and exists in different sports. Elements in this cluster mainly include different groups of people (athletes, children, elites, etc.) and sports performance influencing factors (fatigue, speed, gender, distance, strength, etc.). At the same time, ’swimming’ appears in this cluster, which means that EP exists in this sport. In summary, through the analysis of research trends by keywords, it can be seen that the current research focus are on four aspects: (1) Exploration of clinical medical theories and methods for EP. (2) The objective facts causing EP in overhead sports. (3) Biomechanical study on the joint movement of the elbow joint, or between the elbow joint and shoulder joint, and (4) The effects and relationship between EP and sports performance.

Through analysis of the results of bibliographic coupling of countries and sources, we can catch sight of EP research trends more clearly. Although the USA’s contribution to EP research was the highest, the peak of attention was around the year 2011 or even earlier. A possible reason was that starting pitchers in MLB received astronomically high salaries from 1985 to 2016, which always allowed them to have additional medical services to maintain healthy conditions even though they had the highest chance of elbow joint injury [26]. The research influence of the United States has promoted the exploration of EP in Japan, Britain, Germany, and the Netherlands, and these four countries have the closest connection with the United States in the research of this scoping study. Emerging countries such as China, Spain, and Italy, can be inferred as being at the initial stage of EP research, and their attention comes after 2018. Therefore, the research progress of these three countries on this issue needs more attention and support. The sources the *Journal of Shoulder and Elbow*, *Clinics in Sports Medicine*, *American Journal of Sports Medicine*, *Orthopedic Journal of Sports*, and *Sports Health—A Multidisciplinary Approach* are the journals that contains the most research related to EP, but most of those publications were collected from before the year 2015. However, they had a guiding influence on the most recent publication trends. The *Orthopedic Journal of Sports*, *Sports Health—A Multidisciplinary Approach*, *BMJ Open Sport & Exercise Medicine*, and the *International Journal of Sports Physical Therapy* are the latest journals concerned with EP issues, and they are closely related to the above journals. According to the scope of these latest journals, it can be inferred that the current research on EP is more systematic, the research methods are more novel, and attention has been paid to the research progress in prevention and treatment.

To our surprise, the keyword ’swimming’ appeared in the results of the trend analysis. Although there is little research on EP in swimming at present, it is a positive signal to promote the forward development of EP in overhead sports. A possible reason why scholars began to pay attention to the factor of EP in swimming is that FINA (Fédération Internationale de Natation) has released an athletes’ injury assessment report in this competition cycle after every World Swimming Championships and Summer Olympic Games. This report can guide scholars to pay attention to a specific injury from among the various injuries in swimming and conduct in-depth research to protect athletes’ physical health and prolong their sports careers [27]. Therefore, we believe that the factor of EP in swimming is a new trend in sport science research, and it has great room for growth and value. In the future, once it attracts enough attention from scholars, a large number of studies around it will emerge until completely solved.

Publications on the theme of elbow pain in overhead sports were retrieved from WoS and the data was analyzed objectively and comprehensively. Nonetheless, limitations are still inevitable. Firstly, using specific studies (English, publication type of articles, and review articles) may lead to ignoring the high-quality literature in other languages on the theme of elbow pain and may lead to a biased result. Secondly, citation frequency in the indexed literature is affected by time. Recently published literature may have a low number of citations due to its short time since publication, which leads to differences between research results and the actual situation. Third, only the WoS database was used for the visual analysis, which may lead to a biased picture in results. It is best to add Scopus, PubMed, and other databases for analysis, otherwise many important articles may be excluded.

## 5. Conclusions

Our study proves the increasing research trend on EP in overhead sports. According to the results and discussion, we believe that sport commercialization and the involvement of related professional sports organizations determines the degree of attention to EP in a specific sport and the development of solutions. From our study results, EP in overhead sport in China has aroused great interest despite the number of publications is still scarce and the year of attention being later than 2018. So, the relevant research on this issue in China is worth further exploring. Research results prove that the EP problem exists in swimmers, but the attention to it in related studies is still lacking. Limitations are inevitable, where literature types, database size, and indexing time range can affect the preference for research results. We suggest that a systematic review and meta-analysis of relevant research on EP in overhead sports to be conducted every 3–5 years, and expanding the scope of the database search which can minimize the disadvantages of preference. In the end, based on the research hotspots found in this study, we assume that it will be a valuable research path to continue studying EP’s mechanism, influence, non-clinical action, and relationship with sports performance in specific sports. The exploration of the above research path can be realized, which will form a complete research closed-loop path. In other words, it will be a feasible solution for EP study, which will benefit athletes, coaches, and related professionals in practice.

## Figures and Tables

**Figure 1 healthcare-10-02242-f001:**
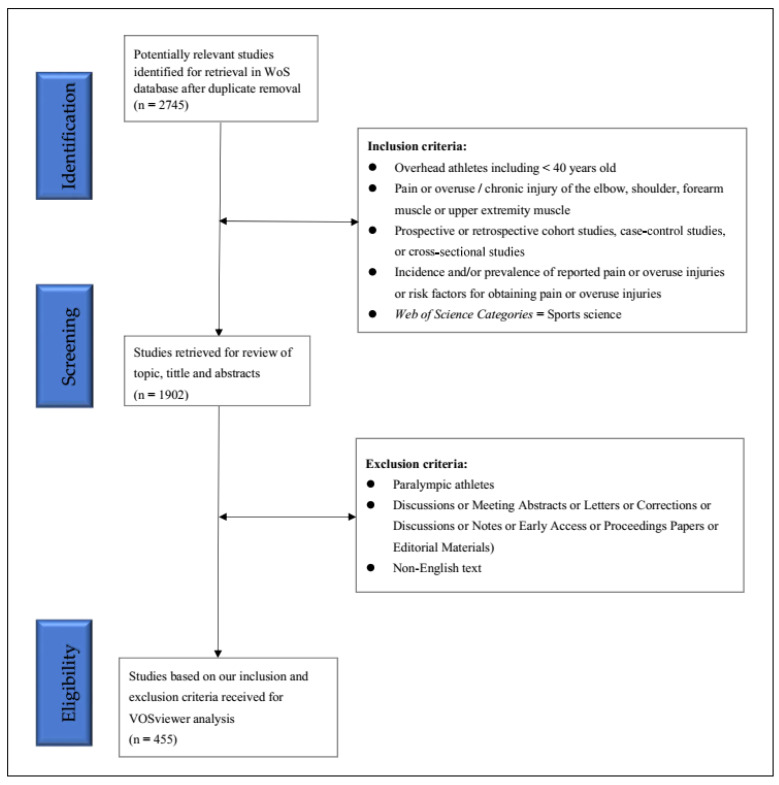
Flowchart of the preferred reporting items for VOSviewer analysis.

**Figure 2 healthcare-10-02242-f002:**
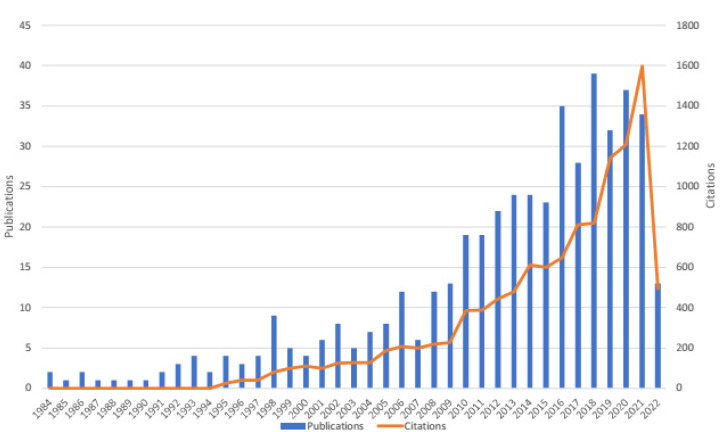
The number of publications and citations per year for 1984–2022.

**Figure 3 healthcare-10-02242-f003:**
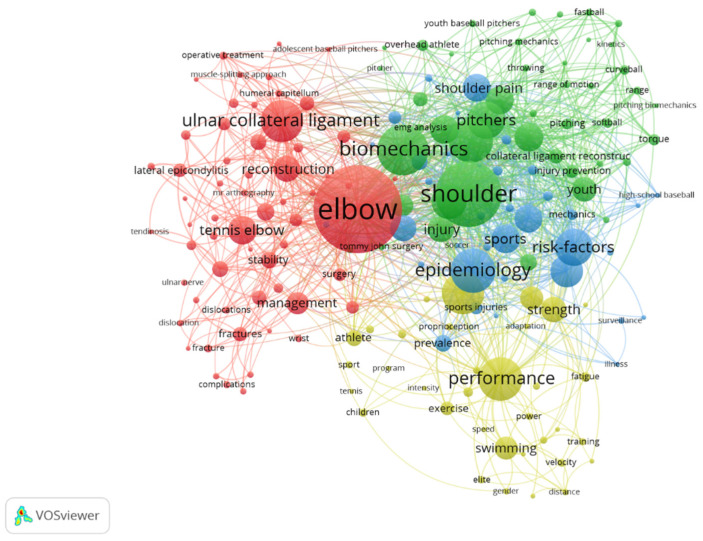
Analysis of the keywords in publications on elbow pain: co-occurrence of keywords. The size of nodes indicates the frequency of occurrence. The curves between the nodes represent their co-occurrence in the same publication. The shorter the distance between two nodes, the larger the number of co-occurrences of the two keywords.

**Figure 4 healthcare-10-02242-f004:**
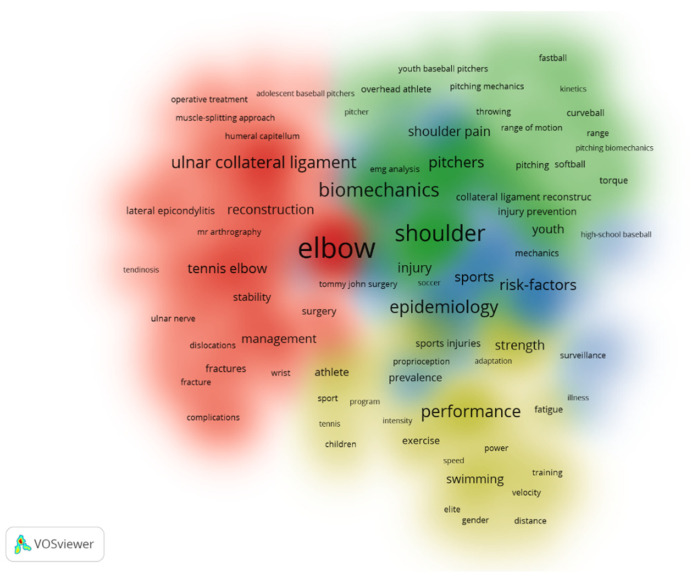
Analysis of the keywords in publications on elbow pain: density visualization. 174 keywords that occurred more than 5 times were enrolled. The font size represents the frequency of occurrence.

**Figure 5 healthcare-10-02242-f005:**
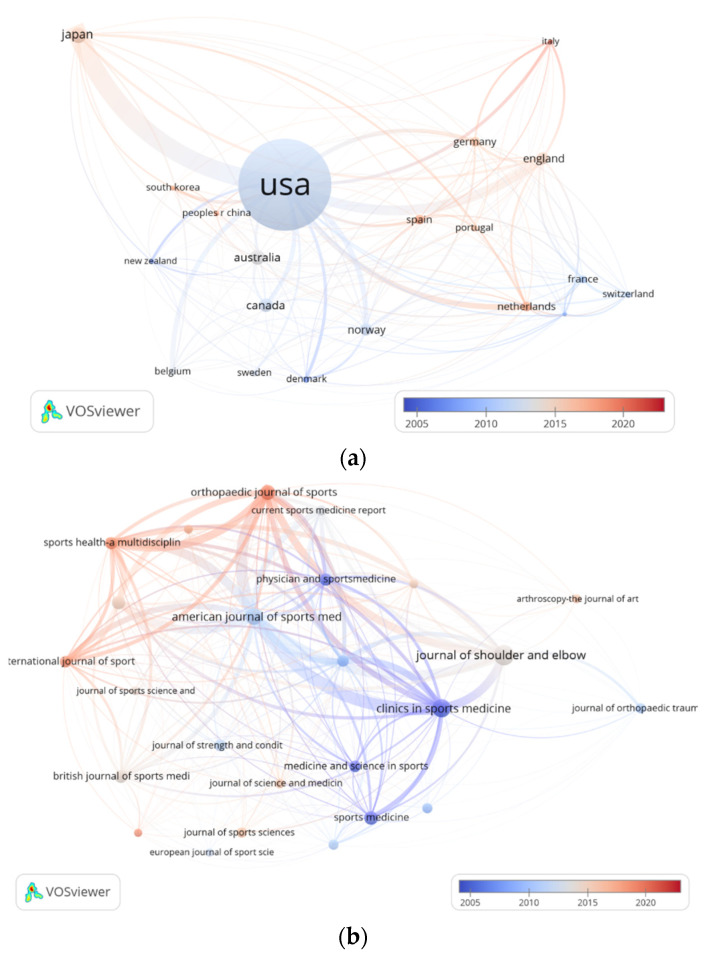
(**a**) Bibliographic coupling of countries. The red color represents the recent trend of country attention to elbow pain. The blue color means that country attention to elbow pain was paid previously. The dimension of the curve indicates the total link strength between countries. (**b**) Bibliographic coupling of sources. The red color represents the recent trend of journal attention to elbow pain. The blue color indicates that journal attention to elbow pain was paid previously. The dimension of the curve indicates the total link strength between journals.

**Figure 6 healthcare-10-02242-f006:**
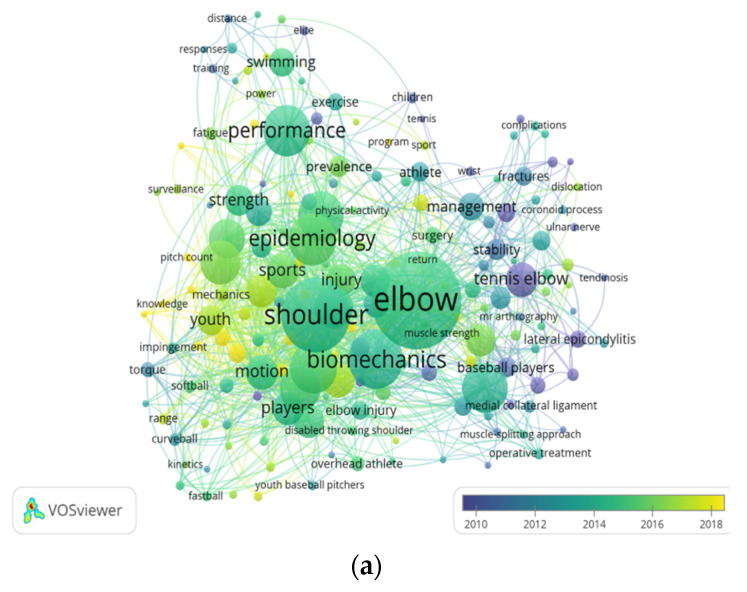
Analysis of themes. (**a**) Network map of trending topics according to the keywords applied from 2010 to 2018. The indicator shows recent publications from green to yellow. More studies focusing on knowledge, pitch count, risk factor, elite swimming, and the prevalence of elbow matters have been published recently. The size of the circles represents the frequency of appearance as keywords. The distance between two circles indicates their correlation. (**b**,**c**) The trend topic of prevalence applied from 2010 to 2018. The indicator shows recent publications from green to yellow. The size of circles represents the frequency of appearance as keywords. The distance between two circles indicates their correlation.

**Table 1 healthcare-10-02242-t001:** (**a**) Top 10 journals of ‘elbow pain’ publications. (**b**) Top 10 organizations of ‘elbow pain‘ publications. (**c**) Top 10 authors of ‘elbow pain’ publications. (**d**) Top 10 countries of ‘elbow pain’ publications.

(a)			(b)		
Journal	Number of Documents	Count of Citations	Organization	Number of Documents	Count of Citations
JOURNAL OF SHOULDER AND 1234567ELBOW SURGERY	50	1011	American Sports Medicine Institute	10	1668
CLINICS IN SPORTS MEDICINE	37	839	Norwegian School of Sports Science	9	594
AMERICAN JOURNAL OF SPORTS MEDICINE	31	2477	Mayo Clinic	9	48
ORTHOPAEDIC JOURNAL OF SPORTS	22	83	Wake Forest University	8	334
SPORTS MEDICINE	18	1863	Hospital Special Surgery	8	175
JOURNAL OF ATHLETIC TRAINING	17	414	Colorado University	8	112
BRITISH JOURNAL OF SPORTS MEDICINE	15	868	Duke University	7	179
SPORTS HEALTH: A MULTIDISCIPLINARY APPROACH	14	360	Oslo University	7	152
PHYSICIAN AND SPORTS MEDICINE	14	94	Columbia University	7	136
MEDICINE AND SCIENCE IN SPORTS & EXERCISE	12	1218	ATI Physical Therapy	6	47
**(c)**			**(d)**		
**Author**	**Number of Documents**	**Count of Citations**	**Country**	**Number of Publications**	**Count of Citations**
ANDREWS, JAMES R	7	1181	USA	226	6530
ANDREWS, JAMES	7	625	Japan	25	367
THIGPEN, CHARLES	6	245	Australia	23	484
SHANLEY, ELLEN	6	244	Canada	21	533
MARINHO, DANIEL ALMEIDA	6	227	England	18	242
ADMAD, CHRISTOPHER	6	138	Norway	18	698
GOODMAN, AVI DELANO	6	44	Spain	14	57
OWENS, BRETT	6	44	Germany	14	120
FLEISIG, GLENN	5	872	Netherlands	14	136
CLARSEN, BENJAMIN	5	510	France	11	172

## Data Availability

Raw and processed data are available upon request to the corresponding author.

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
