# Peer review of "Analysis of Research Trends on Elbow Pain in Overhead Sports: A Bibliometric Study Based on Web of Science Database and VOSviewer"

_healthcare, 2022, doi:10.3390/healthcare10112242_

Round 1

Reviewer 1 Report

I have carefully read the manuscript and my opinion is that the manuscript has a merit to be published in your reputable journal with major corrections. The manuscript is original, informative and readable. The authors tried to find the way how professionals and practitioners can reduce EP risks and support the formulation of suggestions to improve sports performance but also to offer the potential and specific study spot for future research direction. The structure of the abstract should be in the following order: the short explanation followed by purpose of the study, method, results and short discussion with conclusions, so I would appreciate if the authors apply it to be more precisely described. On the other hand, the introduction is well written but I would appreciate if the authors conclude this section with the main aim of the study. Furthermore, I wonder what is the reason the authors did not use the Scopus and PubMed databases in study selection – they would reach mostly the same studies but this database is the most prestigious, just behind the Web of Science. I suggest to retrieve articles via this database searching too. I would like to highlight that it would be very good if this review is registered on PROSPERO, it would show seriousness of the scientific approach.  At the end, I have no amendments on results and discussion part but I would recommend to the authors to prepare the conclusion part in the following order: the main conclusions, the limitations of the study (more precisely) as well as recommendations for the further studies (it is very important to briefly elaborate it and highlight the most important notes). The reference list does not conclude all the studies specified in flowchart, so I wonder if they reviewed all of them or not? Lastly, grammar check and writing style supposed to be checked too. I would recommend you to accept this manuscript right after I confirm the authors revise it in the adequate manner.

Reviewer 2 Report

The authors prepared a well-documented manuscript, congratulations. I have a few suggestions; 

In general, reference style is different throughout the text. The authors used numbers as well as Authors (year), please refer this throughout the text. Also put a space between the sentence and reference numbers.

Introduction was well written explaining why this study is important with relevant previous research.

L43 please use “whose proportions” instead of “which the proportions”

Results

Please use capital letter for journal names

Please pay attention to the writing of the proper names; use capital letters.

Reviewer 3 Report

Dear Authors,

The authors conducted a bibliometric study using VOSviewer as the main tool to identify trends in articles on the topic "Pain in the elbows during overhead sports" and analyzed the main trends associated with this scientific problem. The authors demonstrated the search strategy, and the results obtained. The manuscript is well written and provided with convincing graphs. However, the abstract should be improved in order to more clearly present the methodology and the main results obtained.

My general suggestion is that some minor revision is needed before the manuscript is ready for publication.

Reviewer 4 Report

Dear Authors,

    First of all, I think the manuscript entitled: “Analysis of research trends on Elbow Pain in overhead sports: A Bibliometric Study Based on Web of Science Database and VOSviewer” submitted for publication in the Applied Sciences Journal (MDPI) has a scientific interest.

More specifically:

Ø  The aim of this study is to use VOSviewer as the main tool for identifying trends of the articles on ‘Elbow Pain in overhead sports’ and the type of published articles related to elbow pain and which type of overhead sports were frequently published.

Ø  The paper is well written.

Ø  The text is clear and easy to read [beneficial for this purpose are the targeted and high-definition figures and tables (as well as the appendix material) found within the article].

Ø  The manuscript's conclusions are in accordance with the evidence and the arguments presented by the authors.

Ø  The authors address the central question quite well.

Based on the above:

Overall Recommendation: Accept after minor revision

Comments and Suggestions for Authors:

Lines 110-118: Please add more information and references about how the VOSviewer software (version 1.6.17) works (working methodology).
